# Extreme levels of Canadian wildfire smoke in the stratosphere over central Europe on 21–22 August 2017

Albert Ansmann[1], Holger Baars[1], Alexandra Chudnovsky[2], Ina Mattis[3], Igor Veselovskii[4], Moritz Haarig[1], Patric Seifert[1], Ronny Engelmann[1], and Ulla Wandinger[1]

[1]Leibniz Institute for Tropospheric Research, Leipzig, Germany
[2]Tel Aviv University, Porter School of Earth Sciences and Environment, Tel Aviv, Israel
[3]Observatory Hohenpeissenberg, German Weather Service, Hohenpeissenberg, Germany
[4]Physics Instrumentation Center of General Physics Institute, Moscow, Russia

*Correspondence to:* A. Ansmann
(albert@tropos.de)

**Abstract.** Light extinction coefficients of $500\,\mathrm{Mm}^{-1}$, about 20 times higher than after the Pinatubo volcanic eruptions in 1991, were observed with European Aerosol Research Lidar Network (EARLINET) lidars in the stratosphere over central Europe on 21–22 August 2017. Pronounced smoke layers of 1–2 km vertical extent were found 2–5 km above the local tropopause. Optically dense layers of Canadian wildfire smoke reached central Europe 10 days after injection into the upper troposphere and lower stratosphere caused by rather strong pyrocumulonimbus activity over western Canada. The smoke-related aerosol optical thickness (AOT) was close to 1.0 at 532 nm over Leipzig during the noon hours on 22 August 2017. Smoke particles were found throughout the free troposphere (AOT of 0.3) and in the pronounced 2–km thick stratospheric smoke layer between 14 and 16 km height (AOT of 0.6). The lidar observations indicated peak mass concentrations of 70–100 $\mu\mathrm{g\,m}^{-3}$ in the stratosphere. Besides the lidar profiles, we analyzed Moderate Resolution Imaging Spectroradiometer (MODIS) fire radiative power (FRP) over Canada, and the distribution of MODIS AOT and Ozone Monitoring Instrument (OMI) Aerosol Index across the North Atlantic, showing similar pattern and a clear link between the western Canadian fires and the aerosol load over Europe. We present Aerosol Robotic Network (AERONET) sun photometer observations, compare photometer and lidar-derived AOT, and discuss an obvious bias (too low smoke AOT) in the photometer observations. MODIS AOT for the Leipzig area confirm the high smoke AOT values derived from the lidar measurements. We finally compare the strength of this record-breaking smoke event (in terms of particle extinction coefficient and AOT) with major and moderate volcanic events observed over northern midlatitudes.

## 1   Introduction

Exceptionally dense Canadian wildfire smoke layers causing an aerosol optical thickness (AOT) close to 1.0 at 532 nm crossed central Europe between 3 and 17 km height on 21–22 August 2017. Stratospheric light-extinction coefficients observed at 14–16 km height, about 3–4  km above the local tropopause, were 20 times higher than the maximum extinction values observed in the stratosphere over central Europe in the winters of 1991 and 1992 after the strong Mt. Pinatubo eruption in June 1991

(Ansmann et al., 1997; Jäger, 2005). We never observed such a strong perturbation of stratospheric aerosol conditions with our lidars before and document this record-breaking event in this article. North-American aerosol signatures are usually detected in the height range from 3–8 km over central Europe during the summer seasons (Mattis et al., 2003, 2008) with AOTs of 0.02–0.05, and only in very few cases, enhanced smoke-related extinction coefficients were observed just above the tropopause.

Table 1 gives an overview of extreme and moderate events of stratospheric perturbations (related to volcanic eruptions and wildfire periods) and corroborates the extraordinarily heavy contamination of the lower stratosphere over central Europe on 21–22 August 2017 (more details in Sect. 4).

Record-breaking intensive fires combined with the formation of exceptionally strong and well organized pyrocumulonimbus clusters in western Canada (https://visibleearth.nasa.gov/view.php?id=90759, https://earthdata.nasa.gov/fire-and-smoke-
in-canada) were responsible for these unprecedentedly optically thick stratospheric smoke layers reaching Europe. Fromm et al. (2000, 2003) and Rosenfeld et al. (2007) showed that large amounts of fire smoke can be lifted up to the tropopause within a short time period of less than an hour and partly reach the lower stratosphere via the formation of pyrocumulonimbus clouds which are associated with strong updrafts with vertical wind velocities of 10-30 m/s (Fromm et al., 2010; Peterson et al., 2017). Self-lifting effects (Boers et al., 2010; Siddaway and Petelina, 2011; de Laat et al., 2012) and gravito-photophoresis
forces (Rohatschek, 1996; Pueschel et al., 2000; Cheremisin et al., 2005; Renard et al., 2008) lead to a further ascent of the soot-containing layers.

The 2017 wildfire season (April-September) was the worst ever burning season in British Columbia, Canada, since recording began in 1950 (https://globalnews.ca/news/3675434/2017-officially-b-c-s-worst-ever-wildfire-season/) and exceeds even the year 1958 (855000 ha area burned) with about 900000 ha of burned forest. Recent studies suggest a direct link between
increasing fire activity in northwestern United States and Canada and changing climate and weather conditions (Liu et al., 2009, 2014; Kitzberger et al., 2017). The summer half year of 2017 was unusually dry in western Canada and helped to create a hot, dry environment with a large reservoir of underbrush which is favorable burning material.

According to spaceborne CALIPSO (Cloud-Aerosol Lidar and Infrared Pathfinder Satellite Observation) lidar observations presented by Khaykin et al. (2018) the fire aerosol traveled eastward after entering the upper troposphere and lower strato-
sphere, crossed the North Atlantic, Europe, northern Asia, and circled around the globe within less than 20 days. Khaykin et al. (2018) further pointed out that the optically dense smoke plumes obviously ascended by about 2-3 km per day during the first days after injection into the upper troposphere and lower stratosphere. The smoke layers were observed all over Europe with ground-based lidar systems of the European Aerosol Research Lidar Network (EARLINET) (Pappalardo et al., 2014) and further lidars operated in a long-term mode (Khaykin et al., 2018). Traces of stratospheric smoke were continuously detected
over Europe until the end of 2017.

Large volcanic eruptions were considered for a long time to be the only process by which significant amounts of aerosols and gases can abruptly enter the lower stratosphere. Pyrocumulonimbus-related smoke injections as reported here may foster the discussion on the role and relevance of alternative path ways of massive perturbations of stratospheric aerosol conditions. The extreme August 2017 fire period provides an ideal opportunity to test atmospheric transport models regarding smoke
long-range transport, spread, and removal and the direct and indirect impact of the soot layers on climate. Volcanic and smoke

aerosols show very different chemical, physical and morphological characteristics. In contrast to the liquid and thus spherical sulfuric acid droplets of volcanic origin, stratospheric soot particles are solid, non-spherical, significantly absorb solar radiation, and can ascent to greater heights. Soot particles influence the evolution of ice clouds by serving as ice-nucleating particles in heterogeneous (deposition) freezing processes (Hoose and Möhler, 2012), whereas liquid sulfuric acid droplets influence cirrus

occurrence and evolution via homogeneous ice nucleation (Jensen and Toon, 1992; Sassen et al., 1995; Liu and Penner, 2002). The complex transport features and climatic influences of stratospheric soot layers make it necessary to compare simulated smoke scenarios and the evolution of the smoke layer during long-range transport with available observations. Recent advances in ground-based and spaceborne active and passive remote sensing and thus the availability of a dense set of observations of the biomass burning material will allow detailed model validation studies.

The article is organized as follows: Data analysis and product information are provided next. In Sect. 3 (observations), we begin with an overview of the fire situation in western Canada in August 2017 and the transport of smoke across the Atlantic towards Europe based on Moderate Resolution Imaging Spectroradiometer (MODIS) and Ozone Monitoring Instrument (OMI) measurements. Next, the lidar observations performed from 21-23 August 2017 (episode with maximum stratospheric pollution) are presented. Smoke observations at three lidar stations in central Europe (Leipzig, Hohenpeissenberg near Munich, and

Kosetice near Prague) are shown. Aerosol Robotic Network (AERONET) observations in central Europe and MODIS AOT values around the Leipzig EARLINET/AERONET station will complement the documentation of the extraordinarily strong wildfire smoke event.

## 2   Instruments, data analysis, and products

### 2.1   Lidar data analysis

Three aerosol lidars of the Polly (*Po*rtab*l*e *l*idar *sy*stem) type (Althausen et al., 2009; Engelmann et al., 2016; Baars et al., 2016) were run at the Leipzig EARLINET station (51.3°N, 12.4°E, 110 m a.s.l., Leibniz Institute for Tropospheric Research, TROPOS), at the Hohenpeissenberg EARLINET station (47.8°N, 11.0°E, 1000 m a.s.l., Meteorological Observatory Hohenpeissenberg, German Weather Service), about 60 km southwest of Munich, and at Kosetice (49.6°N, 15.1°E, 500 m a.s.l.), Czech Republic. A triple-wavelength Polly was operated by TROPOS at Kosetice, 75 km southeast of Prague and about

275 km southeast of Leipzig, for three months in the framework of an aerosol field campaign lasting from July to October 2017. During the smoke event northwesterly winds prevailed in the stratosphere and the air masses crossed Leipzig about 3–4 hours before reaching Kosetice.

The so-called Fernald method (Fernald, 1984) was used to derive height profiles of particle extinction coefficient from the lidar observations at daytime. The reference height was set around 10–11 km height (tropopause region). The particle

extinction-to-backscatter ratio (lidar ratio) is needed as input. We used a value of 70 sr for 532 nm. This lidar ratio of 70 sr was measured with our Raman lidars after sunset on 22 August 2017 (Haarig et al., 2018). An uncertainty of the obtained extinction profile is almost directly dependent on the lidar ratio uncertainty. 15% uncertainty in the lidar ratio input parameter (assuming variations of 10 sr around 70 sr) thus leads to a relative uncertainty of about 15% in the smoke extinction values.

To reduce the influence of signal noise the signal profiles have to be smoothed. We used vertical gliding-averaging window length of 185 m in the boundary layer (up to 2.5 km height) and 750 m (above the boundary layer up to 17 km height). The sensitivity tests with different smoothing lengths of 175 m, 350 m, and 750 m in the free troposphere revealed that the main layering features are well resolved by using the comparably large vertical window length of 750 m. The large smoothing length was necessary because the densest smoke layers crossed the lidar at Leipzig during the noon hours when the signal noise by sunlight was the highest.

Temperature and pressure profiles are required in the lidar data analysis to correct for Rayleigh extinction and backscattering. This information is taken from the GDAS (Global Data Assimilation System) data base which contains profiles of temperature and pressure from the National Weather Service's National Centers for Environmental Prediction (NCEP) (GDAS, 2018) with a horizontal resolution of $1°$. We ignore a minor ozone absorption effect at 532 and 607 nm in the determination of smoke extinction coefficient and thus an additional uncertainty of few percent. Alternatively to the GDAS profiles, we selected nearby radiosonde temperature and pressure profiles (Munich, Prague, Lindenberg) in the extinction profile retrieval to check the impact of potential temperature and pressure uncertainties on the results and found rather small deviations between the different particle extinction profiles (of <3% for 22 August 2017). However, one should emphasize that the GDAS data set is based on all available radiosonde observations (in central Europe). The radiosonde profiles are assimilated into the atmospheric model so that the GDAS data (providing temperature and pressure profiles every 3 hours and for distances of typically 20-30 km to the lidar stations) are more representative for the actual meteorological conditions over the lidar sites then the few radiosonde profiles providing the meteorological state for regions typically 60-180 km away from the lidar site and for fixed times (usually for 0 and 12 UTC) and thus coarse temporal resolution.

In Sect. 3, we will also show height-time displays of the volume linear depolarization ratio. This quantity is defined as the ratio of cross-to-co-polarized backscatter coefficient. Co and cross denote the planes of polarization (for which the receiver channels are sensitive) parallel and orthogonal to the plane of linear polarization of the transmitted laser pulses, respectively. The volume linear depolarization ratio is easily obtained from the lidar raw signals and enables us to identify non-spherical particles such as ice crystals and irregularly shaped smoke particles. The depolarization ratio is comparably high when the particles are non-spherical and very low (almost zero) if the particles are spherical (sulfuric acid droplets, soot particles with a liquid shell).

## 2.2 AERONET products

The EARLINET stations at Leipzig and Hohenpeissenberg are collocated with an Aerosol Robotic Network site (Holben et al., 1998). For comparison of the Kosetice lidar observations with respective AERONET measurements we used the data collected at the AERONET Brno site, which is 115 km southeast of Kosetice (and thus downwind of the lidar site at 14–16 km height on 21–22 August 2018). The AERONET sun/sky photometer measures AOT at eight wavelengths from 339 to 1638 nm (AERONET, 2018). Sky radiance observations at four wavelengths complete the AERONET observations. From the spectral AOT distribution for the wavelength range from 440 to 870 nm the wavelength dependence of AOT expressed in terms of the Ångström exponent AE is obtained. Furthermore, the 500 nm fine mode fraction FMF (fraction of 500 nm fine–mode AOT to

total AOT), and particle size distribution for the entire vertical column is derived (O'Neill et al., 2003; Dubovik et al., 2006). Fine mode particles have per definition a diameter of $\leq 1$ $\mu$m.

## 2.3 Satellite-derived products: MODIS and OMI retrievals

Next, we analyzed spatial and temporal pattern of MODIS AOT, MODIS-derived fire radiative power (FRP), and Ozone Monitoring Instrument (OMI) Aerosol Index (AI, 354 nm) observed over the area that covers Western Canada and extends to Europe in August 2017. MODIS AOT values (at 550 nm) were generated by using the GES-DISC Interactive Online Visualization and analysis Infrastructure (GIOVANNI) developed by the National Aeronautics and Space Administration (NASA) Goddard Earth Sciences (GES) Data and Information Services Center (DISC) (Acker and Leptoukh, 2007; Berrick et al., 2009). GIOVANNI provides Level 3 (e.g., 1° spatial resolution pixel size) product that is aggregated and averaged from the Level 2 product (e.g., 0.1° resolution pixel size).

OMI AI for the entire August 2017 period were also produced with GIOVANNI. Positive values of AI are associated with UV absorbing aerosols, mainly mineral dust, smoke and volcanic aerosols. Negative values of AI are associated with non-absorbing aerosols (for example sulfate and sea-salt particles) from both natural and anthropogenic sources (Torres et al., 1998; Buchard et al., 2015; Hammer et al., 2016). Values near zero indicate cloud fields.

The FRP product enables distinction between fires of different strengths at 1 km resolution using Terra and Aqua satellites (Ichoku et al., 2008). Instantaneous FRP values range between 0.02 MW and 1866 MW per 1 km×1 km pixel, with global daily means ranging between 20 and 40 MW (Ichoku et al., 2008). As shown recently by Freeborn et al. (2014), MODIS FRP have an uncertainty of 26.6% at the 1 sigma level. We used active fire products from the MODIS (MCD14DL product) in shapefile formats (https://earthdata.nasa.gov/earth-observation-data/near-real-time/firms/active-fire-data). Only high quality FRP values above 50 MW (and exceeding accuracy levels >65%) were mapped.

Finally, we used 550 nm AOT images for the Leipzig region for 22 August 2017 obtained with the MODIS combined DT (dark target) and DB (deep blue) algorithms (Remer et al., 2013). The most recently released MODIS Collection 6 product MOD04_3K (for Terra) and MYD04_3K (for Aqua) contains AOT at a 3 km horizontal resolution in addition to the L2 10 km product (Remer et al., 2013; Levy et al., 2015). The retrieval algorithm of the higher resolution product is similar to that of the 10 km standard product with several exceptions (for more details, see http://modisatmos.gsfc.nasa.gov/MOD04_L2). Validation against surface sun photometer shows that two-thirds of the 3 km retrievals fall within the expected error on a regional comparison but with a high bias of 0.06 especially over urban surfaces. The uncertainty in the retrieved AOT is 0.05±0.15×AOT for AOT≤1.0 (Levy et al., 2010, 2013). In this study, we use the MODIS Collection 6 (C006) AOT retrievals at 3 km×3 km (at nadir) spatial resolution collected with Terra (10:30 local equatorial crossing time) and Aqua (13:30 local equatorial crossing time) over Leipzig on 22 August 2017 (MODIS, 2018). In addition, we also used Sentinel-2 multi-spectral instrument (MSI) RGB (red green blue) image collected on August 22 to get the cloud cover and ground conditions over Leipzig.

## 3 Observations

### 3.1 Overview: The smoke situation in August 2017 as seen with MODIS and OMI

Figure 1 shows the distribution of fire clusters all over Canada in August 2017 obtained from MODIS observations together with maps of the 354 nm Aerosol Index (OMI) and MODIS-derived 550 nm AOT for the main corridor of smoke transport across the North Atlantic towards Europe. The number of fire pixels with FRP >50 MW (for 1 km× 1 km pixels) was of the order of 10000 in August 2017. The fire activity was highest in British Columbia.

Figures 1b and c show enhanced values of OMI 354 nm AI amd MODIS 550 nm AOT (August 2017 mean values) over the entire region from western Canada to Europe. The spatial features of fire clusters over western Canada match the elevated AOT and AI values over this region. Furthermore, as also evident from Fig. 1, both satellite systems (MODIS, OMI) independently show similar results (pattern) in terms of AI and AOT. The smoke which crossed central Europe on 21–22 August 2017 originated from the most western part of Canada. The source identification aspect is discussed in more detail in Sect. 3.3.

Note that the shown AOT is composed of contributions from aerosol particles in the planetary boundary layer (PBL, about 0.05 over the Atlantic and 0.1-0.3 over the continents), from smoke and anthropogenic haze in the free troposphere (FT, about 0.05-0.25), and from the smoke particles in the stratosphere (S). Tropospheric 550 nm AOT thus ranges from about 0.05 to 0.5 which makes the interpretation of the satellite-based AOT and AI maps concerning the stratospheric aerosol load difficult. The August 2017 mean anthropogenic and marine AOT contribution may have been of the order of 0.1-0.3 at 550 nm. Thus, smoke dominated the AOT pattern in August 2017. Frequently the AOT values exceeded 0.5.

A clear sign for the presence of absorbing wildfire smoke particles are the high AI values. AI frequently exceeded 1.0 and indicates a strongly absorbing aerosol. The impact of anthropogenic haze and marine particles to absorption of radiation at 354 nm is comparably low. AI typically ranges from -0.5 and 0.2 over North America and Europe in the absence of biomass burning (Hammer et al., 2018).

### 3.2 The 21-23 August 2017 smoke event over central Europe

Figure 2 shows the aerosol layering over Kosetice, Czech Republic, from 20–23 August 2017. Coherent smoke structures were observed in the troposphere as well as in the stratosphere over more than one day. Unfortunately, clouds at 1–4 km height disturbed aerosol and cloud profiling considerably during the daytime periods. The tropopause height (indicated as white lines in Fig. 2) was mainly between 10 and 11.5 km from 20–23 August. The extreme event with particle extinction coefficients mostly from 250–500 $Mm^{-1}$ in the lower stratospheric layer lasted from 21 August, 15:00 UTC until 23 August, 5:00 UTC. In the beginning, the 2-km thick smoke layer was detected at 12 km height and at the end at 15–16 km height. Particle extinction coefficients in the free troposphere (between 2-7 km height) were about 20–50 $Mm^{-1}$ on 21–23 August as we will discuss in more detail in Sect. 3.4.

The apparently ascending stratospheric soot layer (observed from 21–23 August) is the results of two different influences. Khaykin et al. (2018) found that the smoke plumes ascended rapidly over the first few days after injection into the upper troposphere with a rate of 2-3 km per day. This cross isentropic ascent was caused by radiative heating of smoke aerosols

(Boers et al., 2010). On the other hand side, the wind velocity decreased with height from the tropopause to 16 km height (GDAS, 2018) as well as from the tropopause to the middle troposphere (5 km height). Caused by the higher velocities in the tropopause region, here first layers crossed the lidar site. The plumes in the middle troposphere and in the stratosphere at 15-16 km height arrived over Kosetice one day later. According to the Prague radiosonde launched on 22 August 12 UTC and

23 August 2017, 0 UTC, wind speeds were of the order of 50 m/s at tropopause levels and about 15-20 m/s at 15–16 km height.

The volume linear depolarization ratio, shown in Fig. 2 (bottom), contains information on particle shape. The depolarization ratio is highest in the cirrus clouds (consisting of strongly light-depolarizing hexagonal ice crystals) and is also significantly enhanced in the stratospheric smoke layer caused by irregularly shaped and most probably dry and non-coated soot particles. Observations (including photographs) of stratospheric smoke particles indicate that stratospheric soot particles can be rather ir-

regular in shape (Strawa et al., 1999). The 532 nm volume linear depolarization ratio was mostly between 0.1 and 0.2 at heights from 12–16 km. The relationship between particle shape and particle linear depolarization ratio is discussed in Haarig et al. (2018).

In contrast to stratospheric particles, tropospheric smoke particles are almost spherical so that the volume depolarization ratio is significantly <0.1. Chemical processing and interaction with particles and trace gases in the troposphere lead to

changes of the shape properties of the fire smoke particles. They are partly coated, embedded, or partially encapsulated after long-range transport (China et al., 2015). Smoke particles with a solid soot kernel and a spherical (liquid) sulfuric acid shell (Dahlkötter et al., 2014) would cause a depolarization ratio close to zero.

### 3.3  Identification of the smoke source regions

The HYSPLIT backward trajectories (Stein et al., 2015; Rolph et al., 2017; HYSPLIT, 2018) in Fig. 3 provide an impression

of the upper tropospheric air flow between North America and central Europe during the 10 days from 12-21 August 2017. According to the backward trajectories, the smoke traveled about 7–10 days from western Canada to central Europe. This is in good agreement with the travel time derived from the spaceborne CALIPSO lidar observation presented by Khaykin et al. (2018).

To identify the sources regions of the wildfire smoke observed over Europe we inspected HYSPLIT forward trajectories

starting over fire region in western Canada. We combined the analysis of forward trajectories with daily maps of OMI AI, MODIS 550 nm AOT and UV Aerosol Index (UVAI) obtained from observations with the spaceborne OPMS (Ozone Mapping and Profiler Suite) as presented in the Supporting Information S2 of Khaykin et al. (2018). Our study was guided by the fact that a rather strong pryocumulonimbus complex (generated from five thunderstorms) developed over the fire areas in southern-central British Columbia (see B.C. region in Fig. 1) and the northwestern United States in the afternoon of 12 August 2017 and

lasted about five hours. It was the biggest pyrocumulonimbus event ever observed, the most significant fire-driven thunderstorm event in history (personal communication, David Peterson, U.S. Naval Research Laboratory, Monterey, California, April 2017). This event obviously triggered the formation of an optically thick smoke layer (probably with an AOT of >2-3) in the upper troposphere and lower stratosphere over British Columbia (see also Hu et al. (2018)). According to the forward HYSPLIT trajectories started over southern-central British Columbia in the afternoon of 12 August 2017 between 5 and 12 km height these

dense smoke plumes traveled northward during the next two days. This is in agreement with the UVAI maps (Khaykin et al., 2018) that show a large region with very high UVAI over northern Canada extending from about 65-75°N and 90–140°W on 14 August 2017 (early afternoon). As further shown in the day-by-day OMPS UVAI maps, the smoke fields then crossed Canada, the Atlantic and split into at least two branches over the eastern Atlantic, and one of these branches crossed central Europe on 21–22 August 2017.

### 3.4 Vertical profiling of smoke

Figure 4 shows the height profiles of particle extinction coefficient at 532 nm as measured over the three Polly lidar sites in southern and central-eastern Germany and Czech Republic on 21–22 August 2017. The locations of the two EARLINET stations at Hohenpeissenberg and Leipzig, and the third one at Kostice are shown in Fig. 5. Maximum extinction coefficients reached 500 $Mm^{-1}$ at all three stations. Values of the stratospheric AOT are given Fig. 5. As mentioned in Sect. 2.1, the uncertainty in the particle extinction coefficients and AOT(s) values is almost directly proportional to the uncertainty in the lidar ratio assumption. We used a smoke lidar ratio of 70 r in the extinction coefficient retrieval (3-17 km height range) as measured with several Raman lidars at Leipzig in the evening of 22 August 2017 (Haarig et al., 2018) and also found over Hohenpeissenberg and Kosetice in the nights from 21-22 August and 22-23 August 2017, respectively. By assuming a realistic smoke lidar ratio from 60–80 sr for 532 nm, the uncertainty in the extinction profiles is of the order of 15%.

The record-breaking smoke event began over Hohenpeissenberg on 21 August (7:00 UTC) and ended on 22 August, 19:00 UTC, and thus occurred 8–10 hours earlier than over Kosetice and 3-5 hours earlier than over Leipzig (upwind of Kosetice). On 21 August, when the first smoke layers arrived over central Europe, the smoke layers were found close to the tropopause. One day later the layers crossed the lidar stations at much greater heights and accumulated in the height range from 14–17 km height, 2–5 km above the local tropopause. These layers traveled with much lower wind speed than the ones at heights close to the tropopause observed one day before. The stratospheric 532 nm AOT in Fig. 5 ranged from 0.37–0.59 during the passage of the densest stratospheric smoke plumes on 21–22 August 2017.

Note that light-extinction coefficients of the order of 500 $Mm^{-1}$ indicate an horizontal visibility of around 6 km in 14–16 km height. At these stratospheric heights the visibility is usually several hundreds of kilometers. Peak mass concentrations were of the order of 70–100 $\mu g\ m^{-3}$ in the lower stratosphere over central Europe around noon of 22 August 2017 (Haarig et al., 2018). Particle extinction values close to 500 $Mm^{-1}$ in combination with lidar ratios around 70 sr corresponds to backscatter ratios (total-to-Rayleigh backscatter) of up to 25-30 at 532 nm at 15 km height. Khaykin et al. (2018) reported backscatter ratios around 10 (ground-based lidar) and almost 20 (CALIPSO lidar) for measurements over southern France in the second half of August and the first half of September 2017.

Figure 6 shows noon and evening lidar profiles for the entire atmosphere over Leipzig from the ground to 16.2 km height. At noon (11:00–12:00 UTC), the entire free troposphere contained traces of smoke. The 532 nm AOT was 0.3 in the free troposphere and about 0.6 in the stratosphere for the height range from the tropopause up to 16.2 km height. The smoke-related AOT was significantly lower in the evening hours (blue curve) with a free tropospheric contribution of 0.08 and a stratospheric

contribution of 0.2–0.25. The evening measurements at Leipzig (22 August, 20:40–23 UTC) with three polarization/Raman lidars are presented and discussed in Haarig et al. (2018).

The high stratospheric AOT of 0.59 over Leipzig is in good agreement with CALIPSO lidar measurements (Khaykin et al., 2018). The maximum AOT measured with the CALIPSO lidar was of the order of 1.0 at 532 nm. These values occurred over northeastern Canada on 17–19 August 2017 and thus a few days upstream of central Europe. Khaykin et al. (2018) originally reported maximum AOTs of 0.7 only, but these values were directly estimated from the height profiles of the attenuated backscatter coefficients. They were not corrected for particle extinction influences (i.e., attenuation effects). If we take smoke extinction (according to an AOT of the order of 0.7–1.0) in the retrieval into account, the true profile of the particle backscatter coefficient multiplied by a smoke lidar ratio of 70 sr leads to an AOT about a factor of 1.5 higher than the apparent one given by Khaykin et al. (2018) and thus to values of the order of 1.0.

### 3.5   AERONET observations at Leipzig

Figure 7 shows the Leipzig AERONET observations from 21–23 August 2017. Level 2.0 data are presented (AERONET, 2018). The lidar observations (diamonds in Fig. 7) conducted between 11:00–12:00 UTC are in good agreement with the extraordinarily high 500 nm AOT of 1.1 at 10:06 UTC. According to our lidar observations, cirrus clouds were absent during the noon hours of 22 August 2017 so that the shown AERONET smoke observations were not affected by any cloud occurrence.

As can be seen in Fig. 7, the 500 nm FMF increased from values below 0.7 in the early morning of 21 August to values close to 1 when the smoke layers arrived and dominated from noon on 21 August to the evening of 22 August. Accordingly, the total AOT was almost equal to the fine-mode AOT caused by anthropogenic haze in the PBL and the smoke in the free troposphere and stratosphere. The Ångström exponent (for the spectral range from 440–870 nm) was mostly between 1.1–1.4 which is indicative for the presence of a pronounced particle accumulation mode (particles with diameter mostly from 200 to 800 nm, see Haarig et al. (2018) for more details).

The boundary-layer 500 nm AOT was around 0.1-0.15 on 21 and 23 August 2017 (before and after the smoke period) and about 0.15-0.2 on 22 August according to the lidar observations at Leipzig. Thus, the fire smoke layers caused a 532 nm AOT close to 1.0 over Leipzig during the noon hours of 22 August 2017.

During the 11:00–12:00 UTC period AERONET 500 nm AOT values ranged from 0.71–0.82 (mean values of 0.76), whereas the lidar-derived 1-hour average 532 nm AOT was 1.1. A realistic lidar ratio of 70 sr was used in the lidar retrieval of the particle extinction profile. This lidar ratio was measured with Raman lidars at Leipzig after sunset on 22 August 2017, but also with the Raman lidar at Hohenpeissenberg (in the night from 21–22 August) and at Kosetice (in the night from 22–23 August). AERONET obviously underestimated the AOT considerably. In Table 2, we summarize several AERONET/lidar comparisons for the three lidar sites used in this study. In all cases, the AERONET 500 nm AOT was significantly lower than the lidar-derived 532 nm AOT. To identify the reason for the bias in Table 2 we provide information about the AOT contributions of the PBL, the free troposphere (FT), and the lower stratosphere (S). As can be seen, the AERONET AOT values are higher than the overall tropospheric AOT contribution. The underestimation of total AOT is thus probably linked to the occurrence of the unusual stratospheric smoke layer.

Strong forward scattering of sun light towards the sun photometer with 1.2° full angle receiver FOV (Holben et al., 1998) seems to be the reason for the underestimation. If small-angle forward scattering is ignored in the AERONET data analysis, the derived (effective) AOT will be much lower than the true (single-scattering-related) AOT when the main aerosol layer (dominating the AOT) is more than 10 km away from the sun photometer. According to Table 2, AERONET underestimated the AOT by about 30–50%.

In contrast to photometers, lidars usually have a very narrow field of view of the order of 0.2–1 mrad (0.01–0.06° full angle) so that forward-scattered laser light does not affect the smoke AOT retrieval. As demonstrated by Wandinger et al. (2010), forward scattering effects only affect spaceborne lidar observations of light extinction in mineral dust layers containing large coarse dust particles, i.e. in cases with strong forward scattering of laser light and a lidar more than 500 km away from the dust layers.

It should be mentioned that small and mesoscale horizontal inhomogeneities in the tropospheric and stratospheric aerosol distributions may have also contributed to the discrepancies between the lidar and sun photometer results in Table 2. Furthermore, a perfect match of lidar and photometer measurement periods was often not possible because of cloud occurrence. In addition, Kostice is 115 km away from the Brno AERONET station so that a direct comparison of Kosetice lidar and Brno photometer observations are not very trustworthy. Finally, the selected smoke lidar ratio of 70 sr may have been too high in

some cases of the lidar extinction retrieval. But all these influences should lead to statistical variations in the lidar-photometer AOT difference around zero, rather than to a clear bias as observed.

### 3.6   MODIS AOT observations over Leipzig

Finally, we analyzed MODIS data around Leipzig. The results are shown in Fig. 8. The MODIS 550 nm AOT values confirm the lidar observation. Many 550 nm AOT values were found above 1.0 during the overflight time (10:10 UTC and 11:55 UTC). The

cloud fields in Fig. 8a provide an impression of the cumulus cloud distribution in the morning of 22 August 2017 (10:10 local time) which hampered the AERONET observations and the MODIS retrieval efforts. Only a few AOT values for 3 km×3 km pixels could be retrieved from the MODIS observations. However, these few AOT values in Figs. 8a and 8b clearly point to AOT values of the order of 1.0 (and higher) at 550 nm in the Leipzig area.

### 4   Discussion

In Sect. 1, we introduced Table 1 to compare the influence of major and moderate volcanic eruptions and extreme and more common pyrocumulonimbus-related biomass-burning events on the aerosol conditions in the lower stratosphere at northern midlatitudes. The goal was to highlight the tremendous contamination of the lower stratosphere with wildfire smoke over central Europe on 21–22 August 2017.

However, there is not doubt that major volcanic eruptions have by far the largest impact on weather and climate. After the

Pinatubo eruption, the sulfuric-acid aerosol was distributed over both hemispheres (Sakai et al., 2016) and the 500 nm AOT at northern mid latitudes was >0.1 for more than two years. Particles were present from the tropopause to about 25 km height for

several years. In contrast, even the extremely large stratospheric smoke contributions caused a mean 532 nm AOT (for the 30-60°N region and for the period from 16 August to 3 September 2017) of the order of 0.01-0.015 only (Khaykin et al., 2018). This AOT is comparable with stratospheric AOT values caused by moderate volcanic eruptions (see Table 1). The vertical extent of the detected smoke layers over Europe in August 2017 was typically <2 km and thus small compared to the vertical extent of the thick Pinatubo aerosol layers of more than 10 km in 1991–1993 (Ansmann et al., 1997).

5 Furthermore, pronounced and dense Pinatubo aerosol layers reached central Europe about 4–6 months after the eruption as a result of the likewise slow meridional air mass transport in the stratosphere from the tropics (<20°N) to lidar sites at >50°N. In contrast, stratospheric particles related to moderate volcanic eruptions and wildfire events at northern midlatitudes are usually advected to central Europe within less than two weeks with the dominating westerly winds (Mattis et al., 2010) so that in these cases almost the maximum impact on the stratospheric aerosol conditions is observable with lidars in Europe. The maximum

10 impact of the Pinatubo aerosol was visible with lidars over Hawaii (DeFoor et al., 1992; Barnes and Hoffmann, 1997) only, about 4–6 weeks after the eruption. The maximum stratospheric Pinatubo-related 550 nm AOT was of the order of 1–1.5 and respective maximum AEC values were of the order of 100–200 $Mm^{-1}$ over tropical regions (Shallcross et al., 2018).

 Nevertheless, the results in Sect. 3 and Table 1 clearly show to what extent wildfires in combination with thunderstorm activity can pollute the lower stratosphere at mid and high northern latitudes and thus can influence radiative transfer, stratospheric

15 circulation and air flow, cirrus formation, and chemical processes. Since the lifting of smoke within convective cumulus towers is so fast (from the fire sources at ground to the upper troposphere and lower stratosphere within <1 hour) (Rosenfeld et al., 2007), and only a minor part of the huge amount of smoke particles can be activated to nucleated cloud droplets at these extremely polluted conditions (personal communication, Daniel Rosenfeld, April 2018), most of the smoke particles reach the tropopause region without any interaction with trace gases, other aerosol particles, and cloud drops. Our lidar profile measure-

20 ments of particle linear depolarization ratio suggest that the majority of the stratospheric smoke particles on 22 August 2017 were uncoated, pure soot particles (Haarig et al., 2018).

 In Table 1, we included the extraordinarily strong pyrocumulonimbus-related Australian wildfire event observed in February 2009 (Siddaway and Petelina, 2011; de Laat et al., 2012). Strong bushfires, very high temperatures, low winds and thunderstorm evolution on 7 February 2009 (Black Saturday) triggered lifting of enormous amounts of smoke towards the upper

25 troposphere from where the smoke layers ascended by the self-lifting mechanism to 15-20 km height (Boers et al., 2010; de Laat et al., 2012). Similar maximum 532 nm AEC and AOT values as measured over central Europe on 21–22 August 2017 were observed with the CALIPSO lidar in the lower stratosphere over the South Pacific east of Australia at heights above the tropopause and below 20 km a few days after 7 February 2009 (de Laat et al., 2012). The stratospheric perturbation slowly decreased during the following months (February-June 2009) (Siddaway and Petelina, 2011).

30 Note finally, that most of the AEC and AOT values in Table 1 are based on standard (elastic-backscatter) lidar observations which enable the retrieval of profiles of the particle backscatter coefficient and the particle backscatter ratio only. A direct measurement of the climate-relevant particle extinction coefficient is not possible with standard lidars. A few Raman lidar studies were available (Ansmann et al., 1997; Mattis et al., 2010) providing direct AEC and AOT measurements as well as measured extinction-to-backscatter ratios (lidar ratios). In Table 1, we used lidar ratios of 25 sr (Pinatubo) (Jäger and Deshler,

2003), 35 sr (moderate volcanic events) (Mattis et al., 2010), and 50 sr during quiet, non-volcanic times (Khaykin et al., 2017) to convert the 532 nm backscatter ratios and backscatter coefficients (derived from the standard backscatter lidar observation) to AEC and AOT values.

## 5    Conclusions

Extreme levels of Canadian fire smoke were observed in the stratosphere, 2–5 km above the local tropopause over central Europe on 21–22 August 2017. Extinction coefficients reached values of 500 $Mm^{-1}$ and were thus about a factor of 20 higher than maximum extinction values found over Germany after the Pinatubo eruption. These rather high stratospheric extinction coefficients were caused by an extraordinarily strong pyrocumulonimbus event over British Columbia in western Canada on 12 August 2017. Several heavy thunderstorms developed over areas with strong wildfires. We analyzed AERONET, MODIS, OMI, and lidar observations to document this historical, record-breaking stratospheric smoke event in terms of 354 nm aerosol index, and 500–500 nm particle extinction coefficients and optical depths. In an accompanying paper (Haarig et al., 2018), we will deepen the smoke characterization towards microphysical, morphological, and composition-related properties based on observations with three polarization/Raman lidar observations at Leipzig after sunset on 22 August 2017.

This extreme stratospheric aerosol event demonstrates that large amounts of wildfire smoke can reach and pollute the lower stratosphere. Such dense smoke layers sensitively disturb chemical processes, radiative fluxes, and even heterogeneous ice formation in the upper troposphere and this probably over weeks to several months. The black carbon aerosol partly enriches the natural soot particle reservoir between 20-30 km by upward motions (Renard et al., 2008).

The unprecedented stratospheric smoke event (observed with EARLINET lidars throughout Europe from mid August 2017 to January-February 2018) provides a favorable opportunity to validate atmospheric circulation models and to improve smoke transport parameterizations. Modeling of the complex life cycle of soot particles (injection, transport, removal by sedimentation, ascent by self-lifting and gravito-photoresis effects) and the complex direct and indirect climatic influences is a challenging effort, but of great importance to improve future-climate predictions and to better understand aerosol-cloud interaction in the upper troposphere. However, high quality and trustworthy modeling is only possible in close connection with vertical profiling of aerosol by means of lidar providing smoke injection heights, smoke burden, size distribution, optical properties, and smoke decay and removal behavior, and all this separately for tropospheric and stratospheric heights.

The spread of smoke was monitored with EARLINET over months with ground-based lidars from northern Norway to Crete and from Evora, Portugal, to Haifa, Israel. Never before, such a dense network of ground-based advanced lidars have been operated in Europe. A systematic analysis of all measurements is planned. Spaceborne CALIPSO and CATS (Cloud Aerosol Transport System, https://cats.gsfc.nasa.gov/data/) lidar observations will be included in the analysis.

A special goal will be the study of the ascent of the soot layers with time. In August 2017, the layers were about 2-4 km above the tropopause, weeks to months later they were mostly observed at heights >20 km, and thus more than 10 km above the tropopause. As mentioned, upward movements of soot containing layers can be the result of heating of the environmental air masses by solar absorption by the soot particles (self-lifting mechanism) and/or by gravito-photoresis effects.

# 6 Data availability

The Polly lidar data are available at TROPOS upon request (info@tropos.de). Backward trajectories analysis has been supported by air mass transport computation with the NOAA (National Oceanic and Atmospheric Administration) HYSPLIT (HYbrid Single-Particle Lagrangian Integrated Trajectory) model (HYSPLIT, 2018) using GDAS meteorological data (Stein et al., 2015; Rolph et al., 2017). AERONET sun photometer AOT data are downloaded from the AERONET web page (AERONET, 2018). We used the ftp site for MODIS data download: https://ladsweb.modaps.eosdis.nasa.gov/allData/6/MOD04_3K/ (MODIS, 2018). OMI AI and MODIS AOT maps (August 2017 mean value) were produced with the GIOVANNI online data system, developed and maintained by the NASA GES DISC (Acker and Leptoukh, 2007). We used active fire products from the MODIS (MCD14DL product) in shapefile formats (https://earthdata.nasa.gov/earth-observation-data/near-real-time/firms/active-fire-data).

*Acknowledgements.* The authors gratefully acknowledge the NOAA Air Resources Laboratory (ARL) for the provision of the HYSPLIT transport and dispersion model. We are also grateful to AERONET for providing high quality sun photometer observations, calibrations, and products. Special thanks to the Lindenberg and Brno AERONET teams to carefully run the stations. We also acknowledge the MODIS mission scientists and associated NASA personnel for the production of the data used in this research effort. This activity is supported by ACTRIS Research Infrastructure (EU H2020-R&I) under grant agreement no. 654109. The development of the lidar inversion algorithm was supported by the Russian Science Foundation (project 16-17-10241).

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

**Table 1.** Comparison of extreme and moderate events of stratospheric aerosol perturbations (volcanic eruptions, pyrocumulonimbus-related smoke events) as observed with lidar over northern midlatitudes. Characteristic values for maximum layer top height (HTOP), typical observable duration of the stratospheric aerosol perturbation (PDUR), maximum 532 nm aerosol extinction coefficient (AEC) and AOT (above the tropopause) are given. For background aerosol conditions, mean AEC and AOT values are shown. The extraordinarily strong (Black Saturday) wildfire smoke event in Australia is included in the comparison. Balloon-borne in situ observations are considered as well (smoke events, background conditions). More explanations are given in Sect. 4.

| Perturbation | HTOP | PDUR | AEC | AOT | Reference |
|---|---|---|---|---|---|
| **Volcanic events** | | | | | |
| Major eruption (Mt. Pinatubo, Philippines, June 1991 ) | 25-35 km | years | 25 Mm$^{-1}$ | 0.2–0.25 | Ansmann et al. (1997); Jäger (2005); Trickl et al. (2013); Sakai et al. (2016); Zuev et al. (2017) |
| Moderate eruption (mid latitudes, tropics) | 15-25 km | months | 5–15 Mm$^{-1}$ | 0.02-0.025 | Mattis et al. (2010); Uchino et al. (2012); Trickl et al. (2013); Sakai et al. (2016); Khaykin et al. (2017); Zuev et al. (2017) |
| Quiet periods | | | 0.2–0.6 Mm$^{-1}$ | 0.004 | Jäger (2005); Khaykin et al. (2017) |
| **Smoke events** | | | | | |
| Extreme case (Canadian fires, Aug. 2017, Australian fires, Feb. 2009) | 15-25 km | months | 500 Mm$^{-1}$ | 0.5-1.0 | this study; Khaykin et al. (2018); Siddaway and Petelina (2011); de Laat et al. (2012) |
| Typical case | 15-20 km | days/weeks | 20-30 Mm$^{-1}$ | 0.02 | Fromm et al. (2010) |
| Background conditions | | | 1.5 Mm$^{-1}$ | 0.005 | Renard et al. (2005, 2008) |

**Table 2.** Comparison of AOT measured with lidar and AERONET sun photometer on 21-22 August 2017. AERONET level 1.0 (Brno), 1.5 (Hohenpeissenberg, 21 August) and 2.0 (Leipzig, and Hohenpeissenberg, 22 August) are used in the table. AOT contributions of the planetary boundary layer (PBL), the free troposphere (FT), and the stratosphere (S) are separately listed in addition. In the lidar retrieval, a lidar ratio of 60 sr (PBL) and 70 sr (FT, S) is used. Brno is 115 km southeast (1.5 hours downwind at 15 km height on 22 August) of the Kosetice lidar site.

| Site and time | PBL | FT | S | AOT(Lidar) | AERONET time | AOT(AERONET) |
|---|---|---|---|---|---|---|
| Hohenpeissenberg (lidar, AERONET) | | | | | | |
| 21 Aug., 13–13:30 UTC | 0.09 | 0.11 | 0.42 | 0.61 | 13:30-16 UTC | 0.32–0.38 |
| 22 Aug., 13–15 UTC | 0.08 | 0.03 | 0.22 | 0.33 | 13–15:30 UTC | 0.23–0.25 |
| Leipzig (lidar, AERONET) | | | | | | |
| 22 Aug., 11–12 UTC | 0.21 | 0.30 | 0.59 | 1.10 | 11–12 UTC | 0.70–0.84 |
| Kosetice (lidar), Brno (AERONET) | | | | | | |
| 22 Aug., 15–15:30 UTC | 0.24 | 0.24 | 0.50 | 0.98 | 13:30–17 UTC | 0.57–0.73 |
| 22 Aug., 18–19 UTC | 0.20 | 0.12 | 0.47 | 0.79 | 17 UTC | 0.65 |
| 22-23 Aug., 22:30–2 UTC | 0.20 | 0.12 | 0.28 | 0.60 | 5-5:15 UTC | 0.45–0.47 |

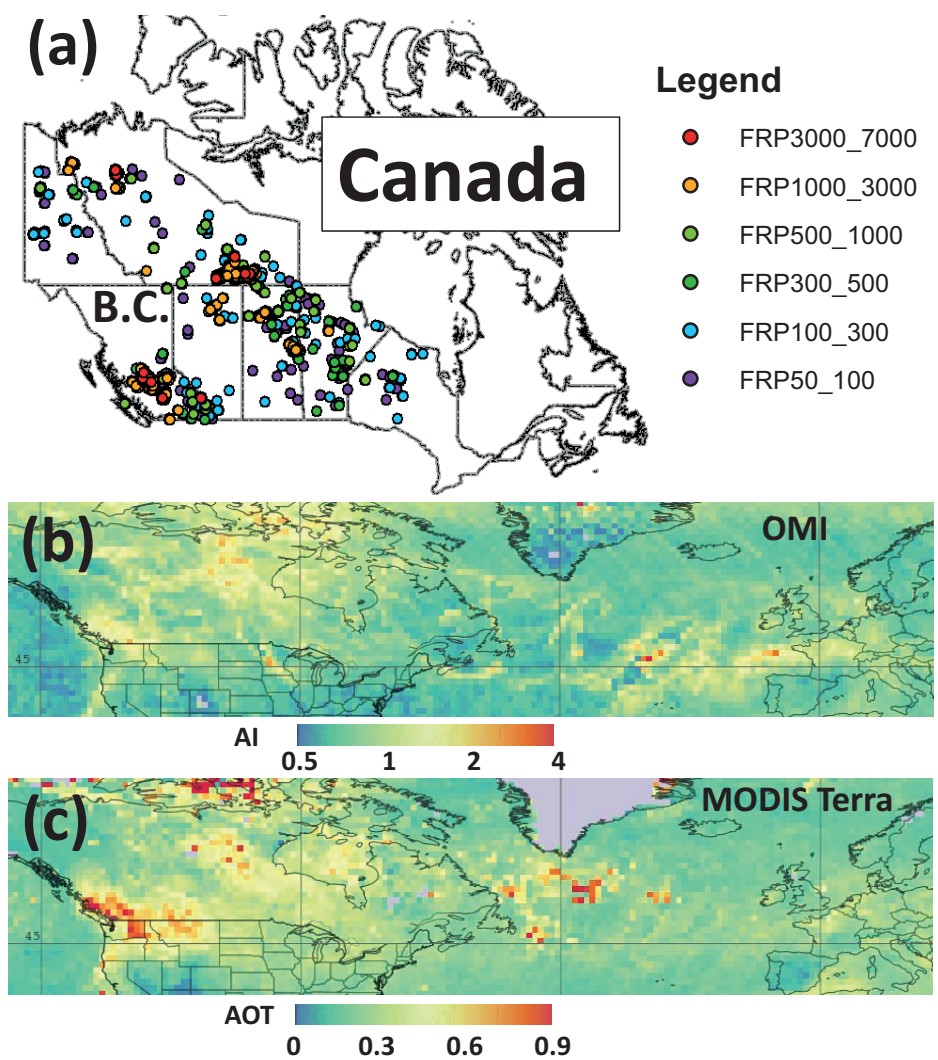

**Figure 1.** (a) Fires detected with MODIS aboard the Terra and Aqua satellites over Canada in the period from 1-31 August 2017. The six color-coded classes of FRP (in MW) indicate different fire strengths (intensity of biomass burning). Intense wildfires accumulated in the southern part of British Columbia (B. C.). (b) August 2017 mean AI (aerosol index, OMI) at 354 nm, and (c) August 2017 mean 550 nm AOT (MODIS) (Acker and Leptoukh, 2007).

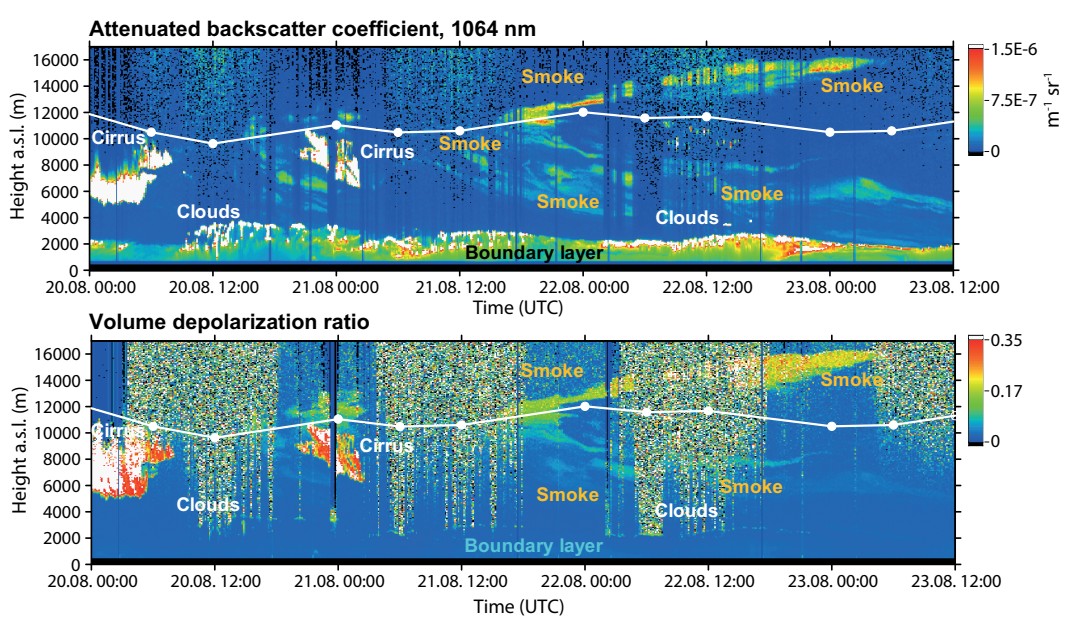

**Figure 2.** Canadian wildfire smoke layers in the troposphere and stratosphere over Kosetice, Czech Republic, observed with lidar on 20–23 August 2017. The uncalibrated attenuated backscatter coefficient (range-corrected signal) at 1064 nm (top) and the volume linear depolarization ratio at 532 nm (bottom) are shown as a function of height above sea level (a.s.l.). Particle extinction coefficients at 532 nm ranged from 250–500 $Mm^{-1}$ in the stratospheric layer from 12 to 16 km height (21 August, 21:00 UTC to 23 August, 5:00 UTC). The tropopause height according to the Prague radiosonde, launched daily at 00:00, 06:00, and 12:00 UTC (see full circles), is given by white lines.

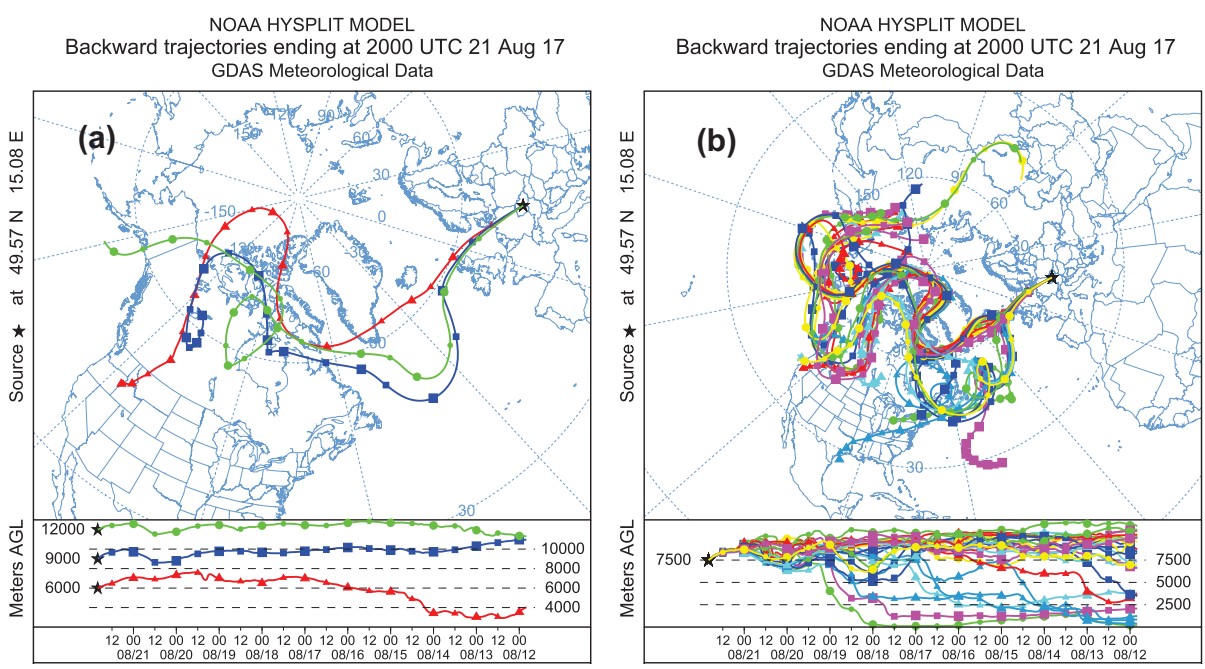

**Figure 3.** 10-day backward trajectories (Stein et al., 2015; HYSPLIT, 2018) arriving at Kosetice, Czech Republic, on 21 August 2017, 20 UTC at (a) 6 km (red), 9 km (blue) and 12 km height (green) above ground level, and (b) ensemble of trajectories for the arrival height of 7.5 km. The trajectories show that the smoke source region is the North American continent.

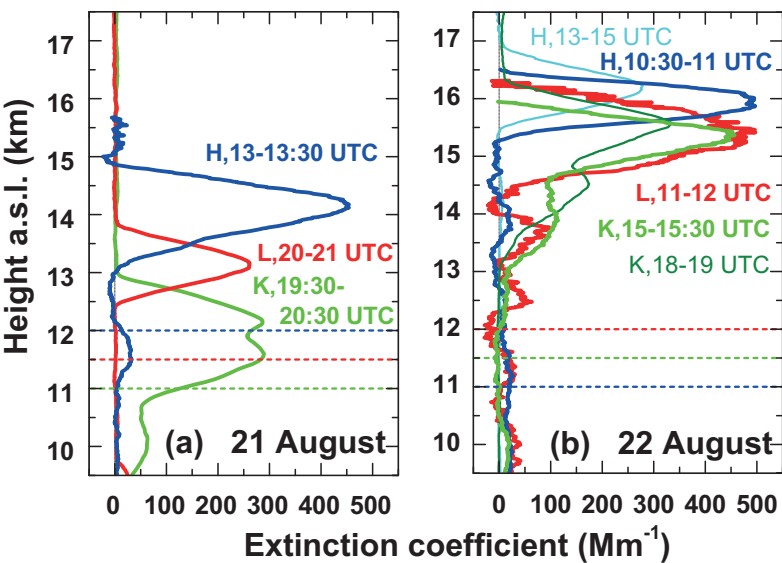

**Figure 4.** Height profile of particle extinction coefficient at 532 nm over Hohenpeissenberg (H), Leipzig (L), and Kosetice (K) on (a) 21 and (b) 22 August 2017. Time periods (in UTC) indicate signal averaging time periods. The Fernald method was applied to compute the extinction profiles. An input lidar ratio of 70 sr (in agreement with nighttime Raman lidar observations of the smoke lidar ratio) was used. The uncertainty in the stratospheric extinction coefficients is estimated to be 15%. The horizontal dashed lines indicate the tropopause heights over the different lidar sites of Hohenpeissenberg (blue), Leipzig (red), and Kosetice (green) estimated from nearby radiosonde temperature and humidity profiles.

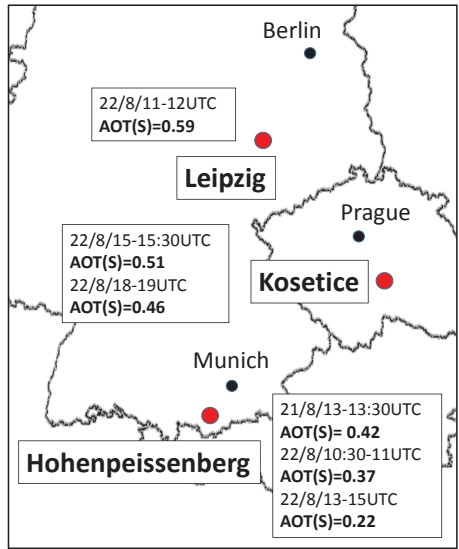

**Figure 5.** Lidar stations (red circles) in Germany and Czech Republic. AOT(S) denotes the AOT at 532 nm of the stratospheric smoke layer as observed on 21 August (21/8) and 22 August 2017 (22/8) for the indicated time periods in UTC.

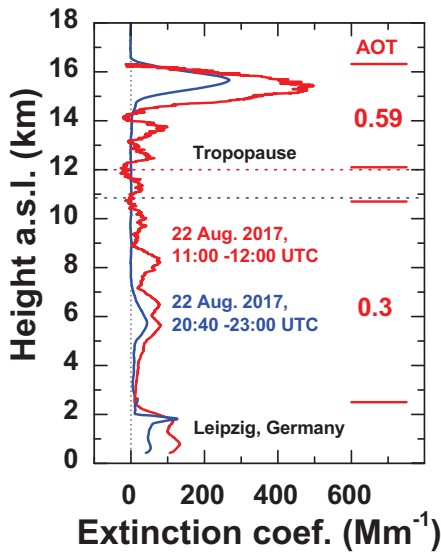

**Figure 6.** Height profile of particle extinction coefficient at 532 nm over Leipzig on 22 August 2017 measured with lidar close to noon (red profile), when the optically densest stratospheric smoke layers crossed the lidar site, and at nighttime (blue profile). The nighttime observations with three lidars will be discussed in detail in Haarig et al. (2018). The data analysis is the same as in Fig 4. The 532 nm AOTs for the free troposphere and lower stratosphere are given as numbers. Horizontal dashed lines indicate the tropopause height around noon and around midnight on 22 August. The uncertainty in the extinction coefficients and AOT values is about 15%.

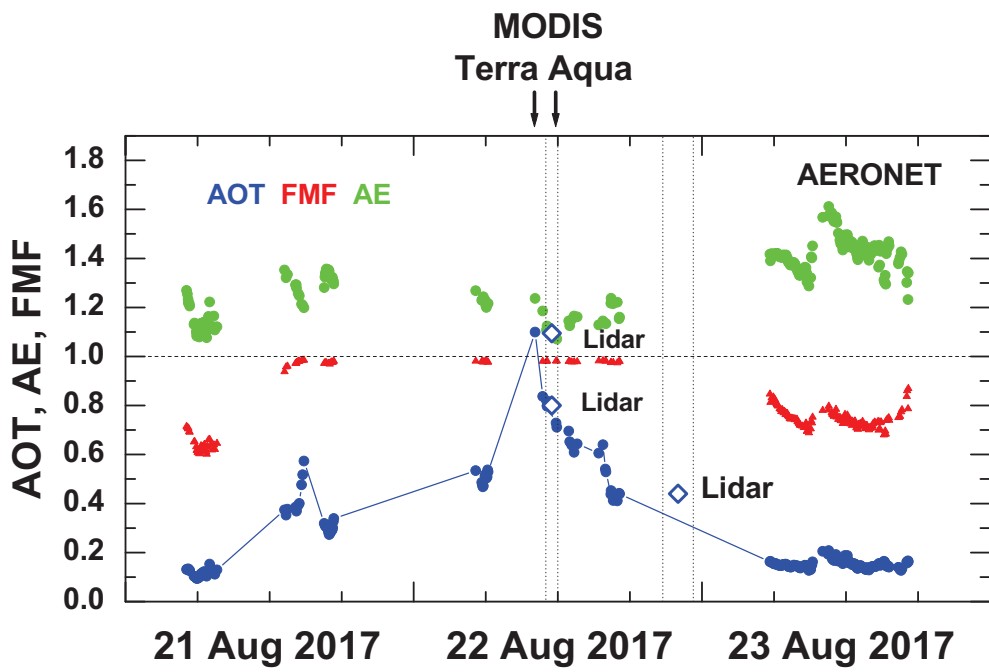

**Figure 7.** AERONET sun photometer observations at Leipzig (TROPOS) from 21–23 August 2017 (level 2.0 data). 500 nm AOT (blue circles), Ångström exponents AE (green circles, for the 440-870 nm wavelength range), and AOT fine-mode fraction FMF (red triangles, for 500 nm AOT) are shown. Gaps in the time series are caused by cloud fields and night time hours. Dashed vertical lines indicate the lidar measurement periods around noon and at nighttime (Fig. 6). The lidar-derived 532 nm AOT (blue open diamonds) are given in addition. AOTs are obtained with an input lidar ratio of 50 sr (AOT of 0.8 on 22 August around noon) and 70 sr (AOT of 1.1 around noon and 0.42 in the night of 22 August). The horizontal line at 1.0 indicates that the FFM values were close to 1.0 during the passage of the smoke layers so that the AOT was almost entirely caused by light extinction by fine mode particles. The overpass times of MODIS Terra and Aqua are indicated above the figure. MODIS results are shown in Fig. 8.

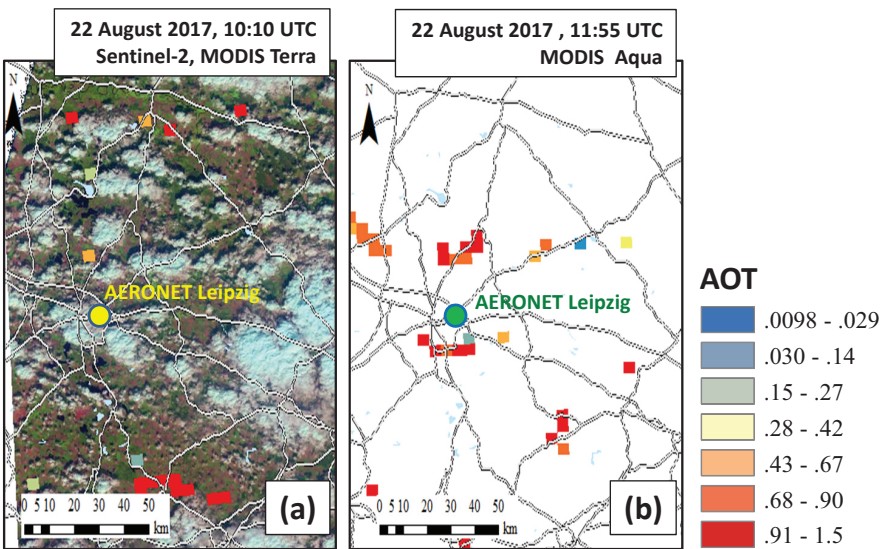

**Figure 8.** (a) Sentinel-2 cumulus cloud fields (bluish-white) and integrated MODIS Terra color-scaled 550 nm AOT values for 3 km×3 km cloud-free areas (Leipzig, overpass at 10:10 UTC), (b) respective MODIS Aqua 550 nm AOT retrievals (Leipzig, overpass at 11:55 UTC). The yellow (a) and green (b) circles indicate the AERONET Leipzig site. Many red squares (AOT of 0.9-1.5) were retrieved from the MODIS observations south and north of the AERONET station.