# Peer review of "Extreme levels of Canadian wildfire smoke in the stratosphere over central Europe on 21–22 August 2017"

_Atmospheric Chemistry and Physics, 2018_

## Referee Comment (RC1) · Anonymous Referee #1 · 4 May 2018

General comments:

This article is an introductory paper (part 1) dealing with the analysis of an extreme event of smoke particles advected from Canada to Central Europe. The work presented here is valuable, especially because high quality and trustworthy climate modeling is only possible in close connection with observations as the ones presented here. In general, the article is writing but my main concern is the reason why this work has been split in two different papers. I recommend merging them into one more robust and complete paper.

Specific comments:

[Figure]

The first part of section "Introduction" seems more like results and you should move to the section "Observations". Because this study is part of the EARLINET network, it would be nice to include a paragraph summarizing the EARLINET findings on this type of particle layers, especially coming from Canada. As suggestion, some these papers are listed below:

Lucja Janicka, Iwona S. Stachlewska, IgorVeselovskii, Holger Baars, Temporal variations in optical and microphysical properties of mineral dust and biomass burning aerosol derived from daytime Raman lidar observations over Warsaw, Poland, Atmospheric Environment, 169, 162-174, 2017.

Ortiz-Amezcua, P., Guerrero-Rascado, J. L., Granados-Muñoz, M. J., Benavent-Oltra, J. A., Böckmann, C., Samaras, S., Stachlewska, I. S., Janicka, Ł., Baars, H., Bohlmann, S., and Alados-Arboledas, L.: Microphysical characterization of long-range transported biomass burning particles from North America at three EARLINET stations, Atmos. Chem. Phys., 17, 5931-5946, https://doi.org/10.5194/acp-17-5931-2017, 2017.

Page 2, line 28. AOT is a quantify depending on the wavelength. It is necessary to specify the wavelength referred to.

Section "Instrumentation" contains more than solely instrumental information. In contrast, methodology details are given here. Please, consider rename this section.

Instrument section should reorganized due to in this paper (part 1) detailed lidar analysis is not the focus. Thus, I recommend to present first Sun-photometer, then MODIS and finally lidar system.

Page 4, lines 21-24. Temperature and pressure profiles needed for the Fernald method were obtained from GDAS. Is there any radiosounding station nearby? Can you quantify the uncertainty introduced by GDAS profiles instead of using actual radiosoundings?

Page 4, lines 26-31. Volume linear depolarization ratio is defined here and used in this

paper. I am wondering why this quantity, which simultaneously provides information of particles and molecules, is preferred instead the particle linear depolarization ratio, which provided information on particles solely.

Page 6, line 15. Here it is stated that the stratospheric smoke particles detected were irregularly shaped. Is it possible to identify the process/processes leading to this kind of shapes using solely your lidar information? Authors refer to the work Haaring et al. (2018). However it would be nice to include some information here.

Figure 7. Which are the error bars associated to these profiles?

Page 8, line 25: Here you present results on particle linear depolarization ratio. However, this quantity has not been defined previously.

Technical corrections:

Page 4, line20: replace "was highest" by "was the highest." Review the entire profile to correct for this typo.

Page 4, line 26: replace "volume depolarization ratio" by "linear volume depolarization ratio". Check the entire body text for replacement.

Page 6, line 25: replace "Figures" by "Figure"

Page 8, line 20: replace "Amercian" by "American"

---

## Referee Comment (RC2) · Anonymous Referee #2 · 12 May 2018

The paper by Albert Ansmann and coauthors documents a record-breaking observation of smoke aerosols above Germany and Czech Republic. The results reported in the Part 1 are based on active and passive remote sensing of aerosol properties using respectively two EARLINET lidars (Leipzig and Kosetice) and two AERONET photometers (Leibzig and Lindenberg). MODIS space-borne observations are used to illustrate the geography of Canadian fires and to provide support for the ground-based observations. The main outcome of the study is based on 3 days of observations and provides estimates of peak extinction coefficient, aerosol optical depth/thickness (AOT), particle mass concentration and accumulation mode effective radius.

[Figure]

The impact of biomass burning and associated pyroconvection on the stratospheric aerosol load is well known. However observational evidence of smoke aerosols in the stratosphere is rare and therefore valuable. The estimates of the smoke plume's optical and microphysical properties reported in the paper are potentially useful for constraining the models, as suggested in the Introduction. The Canadian wildfires during summer 2017 did indeed have an outstanding impact on the stratospheric aerosol and the presented high-quality observations should be documented in the peer-reviewed literature. However, I believe that the study would make a stronger point had it been consolidated with Part 2. Further, the scientific value of this observational study could be much enhanced, if the authors discuss more carefully their observation in the context of other outstanding aerosol events at northern midlatitudes. The comparison of stratospheric impact of the Canadian smoke event with that of Pinatubo tropical eruption is not totally appropriate as explained below. With that, comparisons with midlatitude eruptions and other biomass burning events are totally missing. I suggest that the authors invest an effort towards enhancing the scientific value of this study through consideration of the following remarks.

General remarks.

*It appears that the key statement of this study (as well as of companion paper) is that the observed extinction values were 20 times higher than after Pinatubo. To the casual reader this may suggest that the impact of these fires on stratospheric aerosol is actually much larger than that from the Pinatubo eruption. While this statement is simply misleading, the comparison as such is not correct either. The authors compare peak extinctions of a fresh and compact patch of smoke with that of an older well-mixed volcanic plume spreading over a wide range of altitudes as it was observed by lidars above Europe in 1991-1992. A direct comparison of plumes' optical properties would be justified had a Pinatubo-sized eruption occurred at northern midlatitudes. I believe that before pointing out the superiority of the Canadian smoke peak extinction and AOD over those of Pinatubo, the authors should carefully discuss the aerosol source

locations (upwind at midlatitudes versus tropics) and aerosol transport processes (fast zonal transport within LS jet versus slower meridional exchange and mixing). The temporal extents of the observed stratospheric perturbation due to fires (presented observations cover a few days only) and Pinatubo (several years) should also be discussed. Finally, it would be more pertinent to compare the stratospheric AOD (0.6 vs 0.2-0.3) and not the peak extinctions.

* I am not sure to understand the reasoning for separating this study in two parts. Both parts are centered around lidar soundings and both of them incorporate collocated or nearby AERONET measurements. The distinction seems to be made on the retrieved parameters, e.g. volume particle size distributions in part 1 and mass distribution in Part 2. I think the reader would much appreciate having all the parameters from a time-limited observation in a single article.

* The structural organization of the article should be reconsidered. The introduction appears too lengthy, whereas the discussion section is totally missing. I suggest to introduce the discussion section, which would include some parts of the Introduction and a careful stipulation regarding smoke vs Pinatubo comparison. Section 3 (observations) would be much easier to follow had it been structured by the observation sources.

* Coming back to the extreme extinction and AOD observed by Leipzig lidar. While there is no reason to question the data, I wonder if similar levels of aerosol abundance were observed by the neighboring lidars stations, e.g. Cabauw, Garmish, Hohenpeisenberg, etc. And what is truly puzzling is why the authors do not compare the extinction from two Polly lidars that operated in close vicinity. Is it possible to invert the Kosetice 532 nm data and present the time curtain of extinction rather than attenuated backscatter?

Specific remarks

P.3, The second paragraph should belong to the discussion section

[Figure]

P.4, l.6. Typo in Leipzig longitude

P.5, l.30. "Similar indications . . . were observable. . ." An appropriate reference is missing here.

P.6, l.5. "Figure 3 provides an overview of aerosol layering over Central Europe. . ." To me this is an example rather than an overview.

P.6, l.20-22. Why mention global (general?) circulation models here? HYSPLIT is a Lagrangian trajectory model. Also, how does the pyrocumulonimbus convection influence the long-range transport? I think, the caveats of trajectory analysis for smoke tracking should be explained more carefully.

P.6, l.23. ". . .at tropospheric as well as stratospheric height. . .". The highest-level trajectory is initialized at 12000 m, which is just around the tropopause on 21 August. The smoke was observed as high as 15-16 km above Kosetice so it would be useful to show trajectories initialized at higher levels.

P.7, l.33-34. Here the authors refer to the period of smoke observation by the instruments involved in this study either as the "smoke period" or "smoke event". Such a terminology is not quite correct because the actual period when the smoke was observed above Europe and elsewhere in NH spanned several months.

Figure 1. MODIS detected powerful wildfires at different locations across Canada, particularly in British Columbia and Northwest territories. Which cluster of fires has caused the extreme levels of smoke in the stratosphere? Could anything be inferred from trajectory analysis?

Figure 2. The photos serve a nice illustration of the stratospheric smoke. However, in order to place them in a scientific context, one should at least provide the azimuthal direction at which these photos were taken.

Figure 3. The plot could be much improved by adding the tropopause height curve (or perhaps even a drawn line). This would help distinguishing tropospheric and stratospheric smoke plumes.

[Figure]

---

## Author Comment (AC1) · 14 Jul 2018

**Response to the Reviewers**

The authors are very grateful to the reviewers for their constructive comments. Based on their input, the paper has been significantly revised and improved. The revised version is part of this reply letter.

We avoided to highlight the many, many changes (in color or bold) triggered and recommended by the reviewers to facilitate reading of the revised manuscript.

Let us start with a general remark:

The reviewers suggest to merge the papers: Ansmann et al., 2018 (paper 1) and Haarig et al. (paper 2).

We think this is not a good solution for the following reasons!

Paper #1 (revised version) deals with observation of the record-breaking fire event using spatial technologies and spatial mapping, we investigate the origin of the smoke observed over central Europe and illuminate the transport across the Atlantic, and present time series and maps of AOT observation using different instruments (MODIS, OMI, AERONET, EARLINET lidars). Such events can be hardly monitored by conventional means. Highlight are the height-resolved lidar measurements because only profiling techniques allow us to clearly see and quantify the stratospheric perturbation and the record-breaking contamination. In contrast to paper #1, paper #2 deals mainly with lidar-derived spectral optical, microphysical, and morphoplogical/aerosol shape/type properties of the smoke. So we have two quite different papers.

Paper #1 highlights and documents this historical, record-breaking stratospheric smoke event, whereas paper #2 highlights and focuses on unique lidar aspects, i.e., on smoke profiling with a world-wide unique triple-wavelength polarization/Raman lidar providing lidar ratio and depol ratio profiles at 355, 532, and 1064 nm, and lidar inversion results (size distribution, single scattering albedo, shape characteristics). Such a set of optical profile data has never been presented before.

We would overload the article if we put together the contents of the two papers, even after reduction and condensation of the information content. There are simply too many features and facts worthwhile to be published.

But! We realize that we did a mistake: We selected (almost) the same title for the two quite different papers. The titles are now changed and thus corroborate the difference, and that both papers are stand-alone papers.

Paper #1: Extreme levels of Canadian wildfire smoke in the stratosphere over central Europe on 21-22 August 2017

Paper #2: Depolarization and lidar ratios at 355, 532, and 1064 nm and microphysical properties of aged tropospheric and stratospheric Canadian wildfire smoke

Forced by the comments of the reviewers, we rearranged the papers and fully separated the contents of the two papers, they are practically no longer overlapping regarding the presented results.

**Now to paper #1.**

Our answers are given in bold. The paper is significantly modified and many parts were re-written based on the reviewers comments

A short summary of the main changes in the beginning:

- We added further satellite observations (MODIS 550 nm AOT and OMI 354 nm Aerosol Index from western Canada to Europe) to show the smoke situation (for August 2017) over North America, the North Atlantic, and Europe.

- Therefore, we extended Fig. 1: Besides Fig. 1a (old Fig.1), now we added Fig1b (OMI 354 nm AI map), and Fig.1c (MODIS 550 nm AOT map). We added these figure to highlight the spatial pattern and transport of aerosols to Europe as seen from space. And this event is so unique that we see it.

- We added new Polly observations, performed at Hohenpeissenberg (near Munich). Now we show lidar observations (extinction profiles up to 17 km height) at Hohenpeissenberg, Leipzig, and Kosetice (southeast of Prague) in a new Fig.4 (profiles) and new Fig. 5 (map to indicate the stations, and stratospheric 532 nm AOT over the stations). We extended the studied period (now 21-22 August, not only 22 August).

- We introduced a new section (Sect. 4: Discussion). In Sect. 4, we discuss and compare the properties and features of strong (Pinatubo) and moderate volcanic eruptions and extreme and typical pyroCB smoke events on the stratospheric conditions observable at northern midlatitudes. For this we introduced a new Table 1 showing and summarizing the main features of the different events. Because of Table 1 (and the references mentioned in the table), new literature is added to the references.

- We removed Fig. 2 (photos) and Fig. 8 (a similar figure is discussed in paper #2).

- We moved Fig. 9 (depol ratio spectra comparison, fine-mode smoke vs fine-mod dust) to paper #2.

- We extended the AERONET part (Sect. 3.5). In addition to the presentation and discussion of the Leipzig AERONET observations, we now discuss the obvious bias in the AERONET AOT observations, caused by multiple scattering in case of a sun photometer with a receiver view angle of 1.2°, i.e., 20 mrad. At these unusual conditions with a dense aerosol layer 12-17 km away from the photometer, the effectively measured 500 nm AOT was 30%-50% lower than the 532 nm AOT derived from the lidar measurements. This systematic bias was found for all lidar stations (Leipzig, Hohenpeissenberg, Kosetice/Brno). We summarized the findings in a new Table 2.

Now the step-by-step reply:

**Reviewer #1:**

General comments:

This article is an introductory paper (part 1) dealing with the analysis of an extreme event of smoke particles advected from Canada to Central Europe. The work presented here is valuable, especially because high quality and trustworthy climate modeling is only possible in close connection with observations as the ones presented here. In general, the article is writing but my main concern is the reason why this work has been split in two different papers. I recommend merging them into one more robust and complete paper.

*See our argumentation above. We prefer to keep the two-paper structure, however with practically no overlap (anymore) of the presented findings and results. The papers deal with very different aspects. Based on this comment, the paper was re-written and additional analyses introduced.*

Specific comments:

The first part of section "Introduction" seems more like results and you should move to the section "Observations". Because this study is part of the EARLINET network, it would be nice to include a paragraph summarizing the EARLINET findings on this type of particle layers, especially coming from Canada. As suggestion, some these papers are listed below:

Lucja Janicka, Iwona S. Stachlewska, IgorVeselovskii, Holger Baars, Temporal variations in optical and microphysical properties of mineral dust and biomass burning aerosol derived from daytime Raman lidar observations over Warsaw, Poland, Atmospheric Environment, 169, 162-174, 2017.

Ortiz-Amezcua, P., Guerrero-Rascado, J. L., Granados-Muñoz, M. J., Benavent-Oltra, J. A., Böckmann, C., Samaras, S., Stachlewska, I. S., Janicka, Ł., Baars, H., Bohlmann, S., and Alados-Arboledas, L.: Microphysical characterization of long-range transported biomass burning particles from North America at three EARLINET stations, Atmos. Chem. Phys., 17, 5931-5946, https://doi.org/10.5194/acp-17-5931-2017, 2017.

*We shortened the lengthy introduction (as suggest by reviewer #2). But we would like to mentioned a few results (20 times stronger than Pinatubo) in the beginning of the Introduction to attract readers who know about the major Pinatubo volcanic event. Many papers start in this way.*

*We want to focus on stratospheric smoke only, because only this aspect is very new and provides the record-breaking touch. So, we leave out to discuss the EARLINET findings and observations (they all deal with tropospheric smoke). We would like to emphasize that paper #2 contains an extended literature review with many EARLINET smoke contributions (to lidar ratio and depolarization ratio aerosol characterizations). So, the summary of EARLINET findings is given in paper #2.*

*The two papers (Janicka et al, Ortiz et al.) are nice, relevant, and will be considered now in paper #2. Ortiz et al is already considered in paper #2. But we will add the interesting paper of Janicka et al.*

Page 2, line 28. AOT is a quantify depending on the wavelength. It is necessary to specify the wavelength referred to.

**We agree, and try to mention that we have 500 nm (AERONET), 550 nm (MODIS) and 532 nm (lidar).**

Section "Instrumentation" contains more than solely instrumental information. In contrast, methodology details are given here. Please, consider rename this section. Instrument section should reorganized due to in this paper (part 1) detailed lidar analysis is not the focus. Thus, I recommend to present first Sun-photometer, then MODIS and finally lidar system.

*We changed the title and subtitles of Section 2 accordingly. But we did not change the order of presentation. The paper belongs to the EARLINET special issue. So lidar is most important, and we want to start with lidar. Without lidar, the record-breaking stratospheric event cannot be resolved, cannot be quantified. All the other observations are complementary, but of course very useful and provide the full and consistent picture on temporal and inter-continental scales.*

Page 4, lines 21-24. Temperature and pressure profiles needed for the Fernald method were obtained from GDAS. Is there any radiosounding station nearby? Can you quantify the uncertainty introduced by GDAS profiles instead of using actual radiosoundings?

*Radiosondes are good, but they are only available at certain times. Radiosondes are usually 50-200 km away from the lidar sites. On the other hand, GDAS meteorological fields assimilate all available radiosondes and provide temperature and pressure profiles with a resolution of 1 h and nearest GDAS grid point is usually 20-30 km away from the lidar site. So, meanwhile we believe in GDAS data and prefer to use GDAS profiles in the lidar data analysis. We discuss this aspect now in Sect.2.1.*

*For this specific event (22 August, evening observations) we used Lindenberg radiosonde and GDAS temperature and pressure profiles, the difference (in the extinction profiles) was almost not visible (3% difference, we mention it in Sect 2.1).*

Page 4, lines 26-31. Volume linear depolarization ratio is defined here and used in this paper. I am wondering why this quantity, which simultaneously provides information of particles and molecules, is preferred instead the particle linear depolarization ratio, which provided information on particles solely.

*We use the volume linear depol ratio in Fig.2 (color plot) to identify the stratospheric layer containing mainly irregularly shaped soot particles. The volume depol ratio is easily obtained from the measured signal profiles. Of course, for detailed analysis, we prefer the particle linear depol ratio. But then, the profile of the particle backscatter coefficient has to be computed first, and in the next step, the particle depol ratio can be computed. This is done in paper #2. So, color plots for the particle depol ratio are not so easily obtained, and may contain a lot of uncertainties. Better stay with the color plots for the 'robust' volume linear depolarization ratio.*

Page 6, line 15. Here it is stated that the stratospheric smoke particles detected were irregularly shaped. Is it possible to identify the process/processes leading to this kind of shapes using solely your lidar information? Authors refer to the work Haarig et al. (2018). However it would be nice to include some information here.

*We describe the process in the new section 4 (Discussion). Pyrocumulonimbus convection lifts the smoke within an hour into the stratosphere after Rosenfeld et al. (2007). Most of the smoke is not involved in cloud drop nucleation and will directly reach the stratosphere without any chance to interact with trace gase, aerosol particles and cloud drops. So the fire particles (soot particles) preserve their original shape, and the literature shows us that these soot particles are not spherical and can show complicated shape structures.*

Figure 7. Which are the error bars associated to these profiles?

*We avoid to show error bars, just to avoid to overload the figures. The uncertainties in the extinction profiles are almost linearly connected to the uncertainty in the assumed lidar ratio. We measured lidar ratios around 70 sr, and if the lidar ratio varies from 60 to 80 sr, the uncertainty in the extinction profiles is about 15%. We mention this uncertainty in the figure captions of the figures with extinction profiles (Figs. 4 and 6 and also in Sect.2.1, description of the lidar data analysis).*

Page 8, line 25: Here you present results on particle linear depolarization ratio. However, this quantity has not been defined previously.

*We moved the respective figures and discussions to paper #2.*

Technical corrections:

Page 4, line20: replace "was highest" by "was the highest." Review the entire profile to correct for this typo.

*Done*

Page 4, line 26: replace "volume depolarization ratio" by "linear volume depolarization ratio". Check the entire body text for replacement.

*Agree! … but we use volume linear depolarization ratio (similar to particle linear depolarization ratio) as it is used in the EARLINET community…*

Page 6, line 25: replace "Figures" by "Figure"

*Done*

Page 8, line 20: replace "Amercian" by "American"

*The discussion of the size distribution part (page 8, lines 19-34) is now in paper #2.*

**Reviewer #2:**

The paper by Albert Ansmann and coauthors documents a record-breaking observation of smoke aerosols above Germany and Czech Republic. The results reported in the Part 1 are based on active and passive remote sensing of aerosol properties using respectively two EARLINET lidars (Leipzig and Kosetice) and two AERONET photometers (Leibzig and Lindenberg). MODIS space-borne observations are used to illustrate the geography of Canadian fires and to provide support for the ground-based observations. The main outcome of the study is based on 3 days of observations and provides estimates of peak extinction coefficient, aerosol optical depth/thickness (AOT), particle mass concentration and accumulation mode effective radius.

The impact of biomass burning and associated pyroconvection on the stratospheric aerosol load is well known. However observational evidence of smoke aerosols in the stratosphere is rare and therefore valuable. The estimates of the smoke plume's optical and microphysical properties reported in the paper are potentially useful for constraining the models, as suggested in the Introduction. The Canadian wildfires during summer 2017 did indeed have an outstanding impact on the stratospheric aerosol and the presented high-quality observations should be documented in the peer-reviewed literature. However, I believe that the study would make a stronger point had it been consolidated with Part 2.

*See our argumentation above (at the beginning of this reply letter). There are so many new findings, spectacular results, and unique new measurement approaches that it is impossible to combine so many different measurements and observations showing completely different aspects of this event into a single paper. But, we got the message of the reviewer, and improved significantly paper #1 (see the general remarks in the beginning of this reply letter).*

Further, the scientific value of this observational study could be much enhanced, if the authors discuss more carefully their observation in the context of other outstanding aerosol events at northern midlatitudes. The comparison of stratospheric impact of the Canadian smoke event with that of Pinatubo tropical eruption is not totally appropriate as explained below. With that, comparisons with midlatitude eruptions and other biomass burning events are totally missing. I suggest that the authors invest an effort towards enhancing the scientific value of this study through consideration of the following remarks.

*We agree and followed the suggestion. We introduced section 4 (Discussion) and Table 1 with characteristic numbers (max extinction coefficient, max AOT, duration of the event, depth or top height of the stratospheric aerosol layer) for major and moderate volcanic eruptions, and extreme and more typical pyroCB smoke events. We discuss the differences.*

General remarks.

*It appears that the key statement of this study (as well as of companion paper) is that the observed extinction values were 20 times higher than after Pinatubo. To the casual reader this may suggest that the impact of these fires on stratospheric aerosol is actually much larger than that from the Pinatubo eruption. While this statement is simply misleading, the comparison as such is not correct either. The authors compare peak extinctions of a fresh and compact patch of smoke with that of an older well-mixed volcanic plume spreading over a wide range of altitudes as it was observed by lidars above Europe in 1991-1992. A direct comparison of plumes' optical properties would be justified had a Pinatubo-sized eruption occurred at northern midlatitudes. I believe that before pointing out the superiority of the Canadian smoke peak extinction and AOD over those of Pinatubo, the authors should carefully discuss the aerosol source locations (upwind at midlatitudes versus tropics) and aerosol transport processes (fast zonal transport within LS jet versus slower meridional exchange and mixing). The temporal extents of the observed stratospheric perturbation due to fires (presented observations cover a few days only) and Pinatubo (several years) should also be discussed. Finally, it would be more pertinent to compare the stratospheric AOD (0.6 vs 0.2-0.3) and not the peak extinctions.

*We agree, and introduced Section 4 (Discussion), and discuss all the points mentioned above. We also made an attempt to estimate the maximum Pinatubo effect by looking at lidar data (at Hawaii, DeFoor et al., Barners et al.) and latest global model simulations of Shallcross et al. (these simulation are closely linked to all available lidar observations). The introduced new Table 1 forced us to check all the relevant literature given in the Table.*

* I am not sure to understand the reasoning for separating this study in two parts. Both parts are centered around lidar soundings and both of them incorporate collocated or nearby AERONET measurements. The distinction seems to be made on the retrieved parameters, e.g. volume particle size distributions in part 1 and mass distribution in Part 2. I think the reader would much appreciate having all the parameters from a time-limited observation in a single article.

*We got the message, but please see our argumentation at the beginning of this reply letter. Our mistake was to select almost the same title for both papers and also that we wanted to have a compact set of two papers which are closely related to each other. Now we separated all results, there is almost no overlap between the two papers, and both papers will show up as stand-alone papers with new and unique results worthwhile to be published. I hope this is acceptable!*

* The structural organization of the article should be reconsidered. The introduction appears too lengthy, whereas the discussion section is totally missing. I suggest to introduce the discussion section, which would include some parts of the Introduction and a careful stipulation regarding smoke vs Pinatubo comparison. Section 3 (observations) would be much easier to follow had it been structured by the observation sources.

*This comment triggered a lot. We introduced the requested discussion section (Sect. 4) and Table 1, and also improved the structure of Sect. 3 (observations) by having six subsections (3.1 overview MODIS OMI from Canada to Europe, 3.2 Overview lidar, 21-23 August, aerosol layering, 3.3 source identification, 3.4 lidar profiles, central Europe, three stations, 3.5 AERONET AOT Leipzig and bias discussion, 3.6 MODIS Leipzig area). In subsection 3.5, we have a new Table 2, showing the AERONET AOT vs lidarAOT bias.*

* Coming back to the extreme extinction and AOD observed by Leipzig lidar. While there is no reason to question the data, I wonder if similar levels of aerosol abundance were observed by the neighboring lidars stations, e.g. Cabauw, Garmish, Hohenpeisenberg,
etc. And what is truly puzzling is why the authors do not compare the extinction from two Polly lidars that operated in close vicinity. Is it possible to invert the Kosetice 532 nm data and present the time curtain of extinction rather than attenuated backscatter?

*We added the EARLINET station of Hohenpeissenberg (the German Weather Service has a continuously running Polly lidar), we checked Cabauw and Garmisch, but they have (almost) no data for this time period. The good agreement between the Hohenpeissenberg, Leipzig, and Kosetice observations (especially for the 22 August, see new Fig.4) was surprising. At all three stations, the maximum extinction coefficient was close to 500 Mm-1.*

*We avoid to convert the attenuated backscatter! The attenuated backscatter is easily obtained (range corrected signal at 1064 nm) and nicely shows the evolution of the aerosol layers. If we show the color plot in terms of extinction, the reader may argument that it introduces the high rate of uncertainty. The lidar ratio can be very different in the PBL, free troposphere and stratosphere. Nevertheless, we decided to provide some essential numbers (extinction values) for the pronounced stratospheric and tropospheric structures and layers we found. That is sufficient and helps, we believe.*

Specific remarks

P.3, The second paragraph should belong to the discussion section

*The introduction is completely rewritten and shortened. It is now more compact and straightforward. The second paragraph is removed.*

P.4, l.6. Typo in Leipzig longitude

*Changed*

P.5, l.30. "Similar indications : : : were observable: : :" An appropriate reference is missing here.

*This part is removed.*

P.6, l.5. "Figure 3 provides an overview of aerosol layering over Central Europe: : :" To me this is an example rather than an overview.

*We completely agree with the reviewer. We changed the text accordingly.*

P.6, l.20-22. Why mention global (general?) circulation models here? HYSPLIT is a Lagrangian trajectory model. Also, how does the pyrocumulonimbus convection influence the long-range transport? I think, the caveats of trajectory analysis for smoke tracking should be explained more carefully.

*Unfortunately, we cannot do that. We are not experts for atmospheric modeling (including the HYSPLIT tool). We can show convincing air flow pattern, fortunately for us, but more is not possible. The only trustworthy way to identify the sources of the forest fire smoke is, to our opinion, tracking of aerosol fields by means of satellite data (MODIS, AOT, OMI-AI, OMPS-UVAI) from the fire areas to the lidar sites. We did this in Sect. 3.3: We combined the satellite aerosol features (with rather strong AI) over northern Canada with forward HYSPLIT modeling. Here, we use day-by-day OMPS UVAI maps provided by Khaykin et al. (2018). More details below and in the revised version (see Sect. 3.3).*

*At the end, we found a clear link between the extreme (record-breaking) pyroCB event over southern and central British Columbia on 12 August 2017 and the record-breaking smoke event over central Europe (21-22 August 2017).*

P.6, l.23. ": : :at tropospheric as well as stratospheric height: : :". The highest-level trajectory is initialized at 12000 m, which is just around the tropopause on 21 August. The smoke was observed as high as 15-16 km above Kosetice so it would be useful to show trajectories initialized at higher levels.

*We decided not to show backward trajectories for heights of 15-16 km height. Khaykin et al. (2018) pointed out that the smoke layers ascended by 2-3km per day (cross isentropic transport) during the first 2 days when the AOT was certainly of the order of >2 or even 3 ... probably by the self-lifting effect (triggered by strong absorption of solar radiation). And this aspect is not covered by HYSPLIT, we believe.*

*So, we did not change the trajectory figure (now Fig.3). The main message of the HYSPLIT backward trajectories is: Yes, the smoke-containing air masses we observed at the beginning of the record-breaking episode, 21 August (20 UTC) at heights of 6, 9, and 12 km height above Kosetice originated from North America, roughly from the region of western Canada 10 days before. We are happy that HYSPLIT was able to produce this convincing airflow.*

P.7, l.33-34. Here the authors refer to the period of smoke observation by the instruments involved in this study either as the "smoke period" or "smoke event". Such a terminology is not quite correct because the actual period when the smoke was observed above Europe and elsewhere in NH spanned several months.

*I hope we have now a better definition and more correct terminology. At least we clearly say that we focus on the 'event' or episode which began over the Czech lidar station (Kosetice) on 21 August, 15UTC, and ended on 23 August, 5 UTC. The clear onset and the abrupt end of the record-breaking is another indication that this smoke layer must have something to do with a temporally well-defined meteorological procee (like a thunderstorm). Continuous bush fires would probably lead to more continuous structures in the observed stratospheric aerosol structures. This aspect is also mentioned in Sect.3.3.*

Figure 1. MODIS detected powerful wildfires at different locations across Canada, particularly in British Columbia and Northwest territories. Which cluster of fires has caused the extreme levels of smoke in the stratosphere? Could anything be inferred from trajectory analysis?

*Just to repeat our statements: In section 3.3 (source identification) we explain how we identified the source region (central and southern British Columbia). We used the day-by-day satellite UVAI maps in the Khaykin et al., (2018) paper (14-24 August 2017) and HYSPLIT forward trajectories. We started these trajectories on the afternoon of 12 August when rather strong thunderstorms occurred and formed the strongest pyrocumulonimbus complex ever observed. There is a clear match between these 12 August forward trajectories and areas with rather high UVAI over northern Canada. And these UVAI fields moved then across the Atlantic towards Europe during the next 10 days and one the branches finally crossed central Europe.*

Figure 2. The photos serve a nice illustration of the stratospheric smoke. However, in order to place them in a scientific context, one should at least provide the azimuthal direction at which these photos were taken.

*We removed the figure*

Figure 3. The plot could be much improved by adding the tropopause height curve (or perhaps even a drawn line). This would help distinguishing tropospheric and stratospheric smoke plumes.

*Is now improved, white lines show the tropopause in both plots.*

**Juan- Carlos Antuna:**

General Comments:

The article address a relevant scientific question: the scarce measurements of stratospheric aerosols produced by big wildfire events. The particular event discussed is reported to be the one producing the biggest Aerosol Optical Thickness (AOT) among all the already measured. The measurements from multiple instruments show a noticeable contribution to both the tropospheric and stratospheric aerosols over Europa after its transport from the wildfires region in Canada. The article present novel data measurements but fall short in deriving substantial conclusions. Apart from novelty of the measurements, very few science and discussions is contributed, with the promise to do it in the second part.

*I hope we can convince Juan Carlos that our revised paper was improved significantly. He pushed us forward. Thank you for this! We can say that the paper was largely modified and many parts were re-writen, many new aspects, results, interpretations, and comparisons (Table 1, comparison of volcanic and smoke events) are included in the revised version of paper #1. On the other hand side, to our opinion the fact that we observed a record-breaking stratospheric contamination should be justification enough to publish this observation. We totally agree that papers should contribute to science (better understanding of atmospheric processes), but there must be also room for reporting and documenting spectacular, record-breaking and thus historical events using ground monitoring measurements and space-based observations.*

The reviewer's opinion is that the authors have two options: They may combine parts 1and 2 of the article in one paper or part 1 may be improved and modified considerable. It could be suitable, for example, as an article on Atmospheric Measurement Technique demonstrating how to assemble a set of valuable measurements, including more discussion on the results and highlighting the synergy between the measurements to provide a detailed characterization of stratospheric aerosols originated by wildfire smoke. A detailed discussion will be necessary about the advantages/disadvantages of each instrument with respect to the rest.

*So, we choose the option: improve and modify paper #1 considerably. A discussion on the synergy of different observations, advantages and disadvantages is however not given. We think that it is obvious that many different and complementary observations are always useful. Without different observations the discussion is usually not convincing and trustworthy. It is also obvious that lidar is needed. Otherwise it is impossible the obtain a clear picture of the stratospheric perturbation.*

*We only compare AERONET with lidar results to conclude that AERONET underestimates the AOT at these very unusual conditions with dense aerosol layers in 10 to 17 km distance from the photometer (with receiver full angle of 1.2°, 20 mrad) because of strong forward scattering of sunlight (see Sect. 3.5 and new Table 2 for more details).*

Specific Comments :

The abstract is clear and describe correctly the limited content of the article.

Page 2, line 6: replace "heterogenous" by "heterogeneous"

*We replace all 'heterogenous' in the text…. Now heterogeneous*

Page 3, line 23: replace "ballon" by "balloon"

*We replaced all 'ballon' in the text…. Now balloon…*

Page 4, line 29: replace "shperical" by "spherical"

***Improved throughout the text***

Page 10, line 2: replace "Februray" by "February"

***Improved throughout the text***

The revised version of paper #1 is attached (see next pages)

[revised manuscript text omitted]